# Effect of Chromium and Molybdenum Addition on the Microstructure of In Situ TiC-Reinforced Composite Surface Layers Fabricated on Ductile Cast Iron by Laser Alloying

**DOI:** 10.3390/ma13245750

**Published:** 2020-12-16

**Authors:** Damian Janicki

**Affiliations:** Department of Welding, Silesian University of Technology, Konarskiego 18A, 44-100 Gliwice, Poland; damian.janicki@polsl.pl

**Keywords:** laser surface alloying, ductile cast iron, in situ composite, titanium carbide

## Abstract

In situ TiC-reinforced composite surface layers (TRLs) were produced on a ductile cast iron substrate by laser surface alloying (LA) using pure Ti powder and mixtures of Ti-Cr and Ti-Mo powders. During LA with pure Ti, the intensity of fluid flow in the molten pool, which determines the TRL’s compositional uniformity, and thus Ti content in the alloyed zone, was directly affected by the fraction of synthesized TiC particles in the melt—with increasing the TiC fraction, the convection was gradually reduced. The introduction of additional Cr or Mo powders into the molten pool, due to their beneficial effect on the intensity of the molten pool convection, elevated the Ti concentration in the melt, and, thus, the TiC fraction in the TRL. It was found that the melt enrichment of Cr, in conjunction with non-equilibrium cooling conditions, suppressed the martensitic transformation of the matrix, which lowered the total hardness of the TRL. Moreover, the presence of Cr in the melt (~3 wt%) altered the growth morphology of the synthesized primary TiC precipitates compared with that obtained using pure Ti. The addition of Mo in the melt produced (Ti, Mo)C primary precipitates that exhibited a nonuniform Mo distribution (coring structure). The dissolution of Mo in the primary TiC precipitates did not affect its growth morphology.

## 1. Introduction

Recently, there has been considerable interest in improving the wear properties of machine parts’ working surfaces made from ductile cast iron (DCI) using laser surface treatment methods [1,2,3,4,5,6,7]. A highly effective approach to enhance the wear properties of metallic substrates is the formation of metal matrix composite (MMC) surface layers via laser alloying (LA) [8,9,10]. The wear performance of MMC materials is strongly affected by the morphology, fraction, and types of reinforcing particles, as well as the microstructure of the matrix [11,12,13]. Thus, the processing conditions and the chemical composition of the alloying material are extremely important to obtain the desired wear properties [14]. Depending on the chemistry of the alloying material, LA can produce MMC surface layers on cast iron substrates using both ex-situ and in situ routes [15,16,17]. The in situ MMCs offer the advantage of stronger interfacial bonding between reinforcements and matrix materials than ex-situ MMCs. Moreover, the in situ synthesized reinforcing phases exhibit higher chemical and thermal stability (long-term stability) [18,19]. High C contents in cast iron substrates provide excellent conditions for the in situ formation of MMC surface layers via LA, with elements that tend to form carbides. Several studies on the LA of cast iron substrates have focused on the in situ synthesis of TiC-reinforced layers (TRLs) by introducing Ti into the molten pool [16,20,21,22,23,24,25,26]. From the point of view of the wear properties, the microstructure of the synthesized TRLs should contain a martensitic matrix with homogeneously-dispersed TiC precipitates [24,26]. Moreover, published data on the wear performance of Fe-based MMCs reinforced with TiC phase suggest a significant increase in the wear resistance of TRLs with increasing the TiC fraction [24,26,27,28,29]. However, reported works have shown that during the in situ synthesis of TRLs on cast iron substrates via LA, it is difficult to maintain a homogeneous dispersion of the reinforcing phase upon increasing the Ti concentration in the melt [23]. As a consequence, control over the fraction and growth morphology of the TiC particles, and also the phase composition of the matrix, is highly limited.

Previous studies [23,24,25] were undertaken to determine the feasibility of the in situ synthesis of TRLs on the DCI substrate via LA with pure Ti powder. The aim of that investigation was to establish the processing conditions that allowed the use of the entire C content in the processed substrate to optimize the TRL’s microstructure, i.e., to achieve the highest possible TiC fraction and a fully martensitic matrix. However, the results indicated that a limited amount of Ti can be introduced into the molten pool while maintaining a homogeneous dispersion of the reinforcing phase in the matrix. This limitation was a consequence of a gradual decrease in the intensity of the fluid flow in the molten pool with an increasing TiC fraction in the melt.

In general, the current work extends the concept outlined in earlier works, with an emphasis on assessing the possibility of influencing both the microstructural characteristics of TRL and the intensity of the fluid flow in the molten pool by properly selecting the alloying material chemistry. The primary objective was to determine the effect of the presence of additional Cr and Mo in the molten pool, during the LA of the DCI substrate with Ti, on the growth morphology of TiC particles and also the microstructure of the matrix.

## 2. Materials and Methods 

50 × 50 × 10 mm plates of DCI-grade EN-GJS-700-2 were used as a substrate material (SM). The SM composition, as analyzed by glow discharge optical emission spectrometry, is given in Table 1. The alloying materials were prepared using commercially available powders of pure Ti (99% purity, the particle size range of 45 to 70 µm, H.C. Starck), pure Cr (99% purity, the particle size range of 45 to 70 µm, abcr GmbH), and pure Mo (99.95% purity, the particle size range of 5 to 10 µm, abcr GmbH). Based on unpublished work [30], Ti-Mo and Ti-Cr powder mixtures were prepared in weight percentage ratios of 80–20. The main purpose of the additional introduction of Cr and Mo powders into the molten pool during the synthesis of TRLs was to affect the morphology and composition of the reinforcing phase (TiC precipitates) and also the phase composition of the matrix material. For comparative purposes, the alloying process was also conducted with pure Ti powder.

The LA trials were conducted using an experimental setup with a diode laser (DL020, Rofin-Sinar Laser GmbH, Hamburg, Germany). A detailed description of the experimental setup and the laser processing conditions are given elsewhere [24]. A laser beam spot size (rectangular in shape) was 1.5 × 6.6 mm. During all trials, the laser spot was located on the top surface of SM, and the fast-axis of the laser beam was set parallel to the scanning direction. Based on previous work [23,24,25], the laser power and traverse speed were held constant at 1500 W and 1.25 mm/s, respectively. The powder feed rates depended on the type of the alloying material and are listed in Table 2.

To investigate the influence of the alloying material chemistry on the maximal possible Ti concentration in the molten pool and the compositional homogeneity of the alloyed zone, single-alloyed beads (SAB) were produced (Table 2). For microstructure analysis of the TRLs synthesized at different melt compositions, surface alloyed layers (SAL) were produced via a multi-pass overlapping alloying process with an overlap ratio of 30% (Table 3 and Table 4). Abbreviations used in the manuscript are also listed in Appendix A.

Geometrical parameters of SABs and SALs were measured with an optical microscope (Eclipse MA100, Nikon Corporation, Tokyo, Japan) and image processing software Nikon NIS-EBR (ver. 3.13, Nikon Corporation, Tokyo, Japan).

The microstructure of TRLs was characterized by scanning electron microscopy (SEM) with energy-dispersive spectroscopy (EDS) analysis and X-ray diffraction (XRD). EDS measurements were performed at an accelerating voltage of 15 kV. EDAX Genesis software (ver. 6.0, EDAX Inc., Mahwah, NJ, USA) was used for the collection and analysis of EDS data. The quantitative EDS analysis was made to estimate the concentration of the alloying material in the matrix and also to confirm the compositional homogeneity of the alloyed layers. Additionally, chemical analysis of the SM and SALs was performed on ground surface specimens using a LECO GDS 500A optical emission spectrometer (LECO Corporation, St. Joseph, Michigan, USA). The C concentration in the SM was determined using a LECO CS125 Carbon Analyzer (LECO Corporation, St. Joseph, Michigan, USA). The procedures and equipment used during microstructural characterization are described in greater detail elsewhere [24,25].

Hardness measurements were made on cross-sections of SALs using a Vickers hardness tester (401 MVD, Wilson Wolpert Instruments, Aachen, Germany). Hardness profiles were determined with a 200 g load. The matrix material hardness was measured with a 25 g load applied over 10 s.

## 3. Results and Discussion

### 3.1. Macrostructure Analysis

Figure 1 and Figure 2 present a series of cross-sectional optical macrographs of the SABs produced using Ti-Cr and Ti-Mo powder mixtures respectively, at different powder feed rates. The corresponding cross-sectional areas of the SAB and concentrations of alloying elements in the bead are summarized in Table 2. A detailed description of the effect of the powder feed rate of pure Ti on the shape of the alloyed zone can be found in previous works [23,24,25]. It has been reported that, during synthesis with pure Ti powder, the presence of TiC precipitates in the melt leads to a decrease in the intensity of the fluid flow in the molten pool, which reduces the compositional homogeneity of the alloyed zone [24,25]. Table 2 shows that for SABs produced using a pure Ti, upon increasing the Ti content, i.e., upon increasing the fraction of TiC precipitates in the melt, the fusion area of the SAB decreased. The reduction of the fusion area is, to a certain extent, associated with the fact that the energy of the laser beam melts the alloying powder. On the other hand, the TiC precipitates formed due to an exothermic reaction between Ti and C in the molten pool. Consequently, the higher fraction of TiC precipitates (associated with a higher Ti concentration in the melt) leads to a larger amount of heat generated directly in the molten pool. However, the calculation of the heat generated during the reaction between Ti and C for the TiC content of 15 vol% (SAB no. T2, Table 2) indicated that its value is negligible (~18 J/mm) compared with the heat input of the process (1200 J/mm—the ratio of the laser power and the traverse speed). The above calculations considered the value of enthalpy (Δ*H* = −203 kJ/mol [31]) of the formation of TiC at 2000 °C (the temperature established during examination of the thermal conditions in the molten pool [32]).

Interestingly, the use of a Ti-Cr powder mixture increased the fusion zone upon increasing the amount of alloying material introduced into the molten pool in the optimal powder feed rate range (Table 2). The fusion zone shape of SABs produced with Cr addition indicates that the coefficient of the surface tension temperature on the molten pool surface was positive, producing fluid flow inward along the molten pool surface [33,34,35]. The increase in the powder feed rate increased the fluid flow intensity, leading to a deeper molten pool. In the case of the Ti-Mo mixture, despite an increased powder feed rate, the cross-sectional area of the SAB was generally constant within the optimal powder feed rate range.

From the point of view of shaping the TRL’s microstructure, the use of a complex powder mixture increased the amount of alloying materials in the molten pool at a given heat input (Table 2). This, in turn, significantly increased the Ti content and, thus, the TiC fraction in the uniformly alloyed bead compared with that processed with pure Ti (Table 3). Additionally, the homogeneity of the TiC dispersion throughout the alloyed zone was also improved (Figure 3). Note that, based on the thermodynamic calculations (for 3.6 wt% of C in the DCI substrate), the Ti concentration leading to the formation of a microstructure composed of TiC precipitates embedded in a martensitic matrix was estimated to be in the range of 12 to 13 wt% [24,25]. In the case of SABs made with an 80Ti-20Cr powder mixture, the Ti content reached 11.6 wt% (SAB no. TC3, Table 2). The multi-pass overlapping alloying process under the above processing conditions using an overlap ratio of 30% ensured the production of homogenous SALs (TRC, Table 3) with average Ti and Cr contents of 12.4 and 2.9 wt% respectively, and a uniform thickness of ~1.9 mm (Figure 4a). In turn, the use of the 80Ti-20Mo powder mixture produced a uniform SAB containing up to 9.9 wt% Ti and 2.8 wt% Mo (SAB no. TM3, Table 2). The corresponding SALs (TRM, Table 3) produced at an overlap ratio of 30% contained about 10.5 wt% Ti and 3.1 wt% Mo. The thickness of this SAL was about 1.6 mm (Figure 4b).

Regardless of the used alloying material, both SABs and SALs contained cracks induced due to a lack of preheating the substrate. The chemistry of alloying material had little effect on the cracking tendency of the alloyed zone. All types of SALs exhibited minor porosity and negligible tendency to a formation of microvoids.

It should be pointed out that the research program adopted in the presented work was not directly aimed at fully understanding the above effect of Cr and Mo on the heat and fluid flow phenomena in the molten pool during the investigated LA process. The complexity of the heat and mass transfer phenomena in the molten pool, especially comprising the significant changes in the surface and bulk chemical composition of the molten pool, requires a detailed examination using both experimental and numerical approaches. However, the results of the performed research revealed a possible factor that may affect the observed changes in the intensity of the fluid flow in the molten pool.

It is well known that mass transport in the molten pool can be substantially altered by the presence of surface-active trace elements such as sulfur and oxygen. Additionally, it is well documented [33,34] that in the case of Fe-based alloys, surface-active trace elements affect the molten pool shape by altering surface tension gradients, leading to changes in both the magnitude of fluid flow and its direction in the molten pool. The chemical analysis presented in Table 4 revealed a significantly elevated S content in all SALs compared with that of the as-received DCI (Table 1). The S content in the as-received DCI was approximately 0.005 wt%. In the case of the SALs produced with added pure Ti powder, the S content gradually increased upon increasing the Ti concentration [25], i.e., with increasing the powder feed rate, and reached 0.148 wt%. The SALs produced using powder mixtures of 80Ti-20Cr and 80Ti-20Mo under the optimal processing parameters exhibited essentially the same sulfur content of 0.180 wt%. Thus, the elevated S content in all types of SALs was associated with the purity of the alloying powders. Based upon investigations on the heat transport in the molten pool [35], which indicate that small concentrations of surface-active elements can significantly alter the magnitude of surface tension, it is suggested that the above differences in the S content may affect the intensity of fluid flow in the molten pool.

Summarizing the above results, the introduction of additional Cr or Mo powder into the molten pool during the LA of the DCI with Ti powder enables significantly higher Ti concentrations in the uniform SAL relative to that achieved using a pure Ti powder by influencing the intensity of fluid flow in the molten pool. However, a full understanding of this phenomenon requires additional examinations and presents a challenge for further research.

### 3.2. Microstructural Analysis

BSE SEM micrographs taken from different locations of the TRC and TRM are presented in Figure 5 and Figure 6, respectively. Representative XRD patterns for TRC and TRM are presented in Figure 7a,b, respectively. A detailed microstructural characteristic of TRL synthesized during LA with a pure Ti powder is presented in previous works [23,24,25]. The chemical compositions and microstructural parameters of the TRLs produced using all types of alloying materials are outlined in Table 3 and Table 4, respectively.

The micrograph analysis showed that the TRC and TRM exhibit a highly uniform distribution of TiC precipitates throughout the matrix and have similar fractions of about 21 vol%. This TiC fraction is close to that predicted by the thermodynamic calculations for the processed DCI substrate [24,25]. As a result, both the TRC and TRM contained minor fractions of the cementite phase (~2 vol%). Moreover, in contrast to the TRLs formed with pure Ti [25], there is no evidence of graphitization in the heat-affected zone (HAZ) between consecutive beads in SALs (Figure 5b and Figure 6d). As can be seen from Figure 5c and Figure 6c, the obtained TRC and TRM compositions also led to the formation of fine eutectic precipitates of the TiC phase with a plate-like morphology.

The microstructure of the TRC also contained a minor amount of fine eutectic-like regions that exhibited two discrete structures (Figure 8a). The corresponding SEM elemental mapping (Figure 8c–g) indicated C enrichment in both structures, wherein that containing remarkably larger lamellar precipitates was also significantly enriched in Cr. In contrast, the structures composed of fibrous-like precipitates (Figure 8b) were depleted in Cr and enriched in Si. Moreover, the elemental maps of C and Ti clearly suggest the presence of TiC precipitates within this eutectic-like region. The above eutectic-like regions formed as a result of the last solidification stages in the Fe-C-Ti-Cr alloy system taking place under non-equilibrium conditions. It is reasonable to assume that the lamellar precipitates enriched in Cr were (Fe, Cr)_3_C phase. Quantitative analysis of the micrographs indicated an average volume fraction of (Fe, Cr)_3_C precipitates in the TRC of 2.1% ± 0.6%. (Fe, Cr)_3_C phase was not detected by the XRD due to the low fraction of this phase.

The typical eutectic-like structure formed during the last stages of solidification in TRM is presented in Figure 9. The fraction of these eutectic regions in the layer was estimated to be 1.9 ± 0.6 vol%. As in the case of (Fe, Cr)_3_C phase in TRC, the XRD did not detect the presence of these eutectic precipitates. However, considering the morphology of the above eutectic-like structure and the chemistry of the layer, one can assume that it contained mainly the cementite phase.

The metallographic data indicate that the addition of Cr significantly elevated the retained austenite fraction in the matrix material (Table 3). The fraction of retained austenite in the TRC was estimated to be 24 wt%, whereas the TRM had austenite fractions of about 5.7 wt%. The overall high retained austenite fraction in the TRC was directly attributed to the combined effect of non-equilibrium cooling conditions in the molten pool and enrichment of primary austenite grains in Cr, which inhibited the martensitic transformation. Based on the SEM/EDS analysis, the Cr content in the martensitic/austenitic matrix was approximately 2.4 wt%. The suppressed martensitic transformation in the laser-processed SALs on the DCI substrate by relatively low Cr additions has been reported in the previous work [36].

The TRC exhibited a non-uniform distribution of retained austenite throughout the layer because of the overlap reheating during multi-pass alloying (Figure 5a,b). Metallographic data suggest that, in the HAZ between consecutive beads in the TRC, a significant retained austenite fraction was converted to martensite due to reheating in this region. This is supported by the hardness distribution in the layer discussed in the next section. In turn, the microstructural characteristics of the TRM clearly show a Mo concentration of about 3.1 wt% in the molten pool during the LA, which Ti provides a completely martensitic matrix, as desired. Furthermore, minor microstructural changes in the overlap boundary (not a significantly elevated austenite fraction) indicates a notable improvement in the thermal stability of the TRM matrix compared with that of the TRC.

Note that in the case of the cubic MC carbide, Mo is a reactive element, i.e., Mo atoms can substitute for Ti atoms to form (Ti, Mo)C phase [37]. Thus, the addition of Mo into the melt provided an opportunity to change the composition of the synthesized cubic MC (TiC) carbide and led to the formation of primary and eutectic (Ti, Mo)C precipitates (Figure 10). It should be pointed out the SEM/EDS analysis detected negligible Mo content in the TRM matrix material. This indicates that the entire amount of Mo introduced into the molten pool dissolved in the TiC phase. Thus, considering the total concentration of MC carbide-forming elements in the melt (~10.5 wt% of Ti and ~3.1 wt% of Mo), it can be concluded that the TRM composition was optimal from a thermodynamics perspective, i.e., required to achieve the highest possible TiC fraction and a fully martensitic matrix [24,25]. Primary (Ti, Mo)C precipitates were observed in the compositional gradient between the core and outer layer (coring). The SEM/EDS elemental mapping, presented in Figure 10, suggests that the center of the primary dendritic (Ti, Mo)C precipitates was depleted in Mo, whereas the outer regions solidified with progressively higher Mo concentrations. A detailed characterization of the primary (Ti, Mo)C precipitates is presented elsewhere [38].

As expected, based on reported data [39,40], Cr addition did not affect the composition of the synthesized TiC precipitates. The EDS elemental mappings presented in Figure 11 indicted homogenous compositions of the TiC precipitates in TRC. Figure 11 also shows a distribution of the previously mentioned Cr-rich eutectic regions in the martensitic/austenitic matrix of TRC.

Quantitative metallography revealed that, despite the same fractions of primary TiC/(Ti, Mo)C precipitates, TRC exhibited a slightly lower mean free path (MFP) than TRM. The MFP of TRC and TRM were 15.1 ± 2.4 and 17.6 ± 3.4 µm respectively, which was directly attributed to differences in the morphology between dendritic precipitates of primary TiC and (Ti, Mo)C phase. It is well-established that the MFP of composites depends on both the fraction and morphology of the reinforcing particles [41,42]. Comparing the primary TiC precipitates in TRC (Figure 5, Figure 11a and Figure 12a) and (Ti, Mo)C in TRM (Figure 6, Figure 10a and Figure 12b) shows that the difference in the composition of the molten pool affected the growth morphology of the primary TiC/(Ti, Mo)C phase. The TiC precipitates in TRC have markedly thinner primary and secondary arms compared with (Ti, Mo)C in TRM. For comparative purposes, Figure 13 presents an SEM image taken of the TRL processed with pure Ti, which was non-uniformly alloyed, but the analyzed region contained approximately 12.5 wt% Ti (the same Ti content as in the TRC). This micrograph generally showed the same growth morphology of the primary TiC phase as (Ti, Mo)C, proving that Mo addition did not affect the morphology of the primary (Ti, Mo)C phase in the TRM. Thus, it is reasonable to suggest that Cr addition during the investigated LA process influenced the nucleation or growth mechanism of crystals of the primary TiC phase. It was reported in several works [43,44,45,46,47,48] that the growth morphology of TiC crystals during the melt reaction method strongly depends on the composition of the melt environment.

### 3.3. Hardness Analysis

Figure 14a,b shows typical hardness depth profiles of TRC and TRM, respectively. The corresponding hardness profiles across the overlap boundary in these layers are presented in Figure 14c,d. In general, due to the minor fraction of retained austenite, the TRM exhibited a higher overall hardness (610 ± 20 HV0.2) than the TRC (545 ± 25 HV0.2). Moreover, the Mo addition during the investigated LA process provided an almost homogeneous hardness distribution in the layer, even at the overlap boundary region (Figure 14d). The hardness uniformity of the TRM confirms that the addition of Mo produced a matrix material with high thermal stability. In the case of the TRC, as previously mentioned in the microstructure section, Cr addition leads to a relatively high fraction of retained austenite that, in turn, undergoes martensitic transformation in the HAZ of the overlap boundary, resulting in large hardness variations in the layer. The average hardness of the matrix in the TRM was estimated to be 580 ± 34 HV0.025. In the case of TRC, the matrix hardness ranged from 480 ± 32 to 550 ± 41 HV0.025 near the center of a single processed bead and in the HAZ of the overlap boundary, respectively.

## 4. Conclusions

This manuscript presented a novel approach to the in situ synthesis of TiC-reinforced composite surface layers on DCI substrates via LA that, by properly selecting the molten pool composition, changed the characteristics of TiC precipitates, the phase composition of the matrix material, and also the dispersion uniformity of TiC phase. In contrast to alloying with pure Ti, using 80Ti-20Cr and 80Ti-20Mo (weight ratio) powder mixtures produced the optimal TiC fraction from a thermodynamics perspective, i.e., the highest possible TiC fraction for the C content in the processed DCI [24]. As a result, the developed approach makes it possible to convert an as-cast structure into a composite structure containing TiC precipitates uniformly dispersed in the completely metallic matrix (eutectic cementite was limited to ~2 vol%). However, the presence of additional Cr in the molten pool, in conjunction with non-equilibrium cooling conditions, suppressed the martensitic transformation of the matrix material, leading to an undesired elevated amount of retained austenite (~24 wt%), and, consequently, a low matrix hardness. Additionally, the elevated retained austenite fraction in the matrix significantly changed the hardness in the overlap boundary in the SAL. In contrast, the addition of Mo to the melt led to the formation of an almost fully martensitic matrix (~5.7 wt% of retained austenite fraction), ensuring a uniform hardness distribution throughout the SAL.

It was found that the introduction of Mo into the molten pool led to the synthesis of primary and eutectic precipitates of (Ti, Mo)C phase. The primary (Ti, Mo)C precipitates exhibited a nonuniform Mo distribution (coring structure). The growth morphology of the primary (Ti, Mo)C precipitates was similar to that of TiC precipitates synthesized during processing using pure Ti. The presence of additional Cr in the molten pool did not affect the composition of the primary TiC phase. However, the melt enrichment of Cr (~3 wt%) refined the primary dendritic precipitates of TiC phase compared with that formed during alloying with a pure Ti and Ti-Mo powder mixture. As a result, at a constant fraction of TiC precipitates, the mean free path between the reinforcing precipitates was reduced.

## Figures and Tables

**Figure 1 materials-13-05750-f001:**
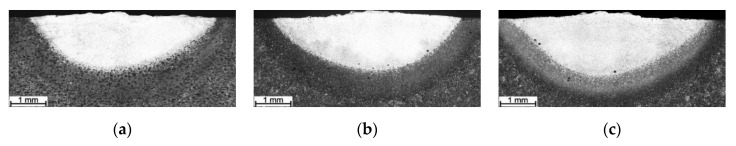
Optical macrographs of the SABs produced using a powder mixture of 80Ti-20Cr (weight ratio); SAB no. (Table 2): (**a**) TC1, (**b**) TC2, (**c**) TC3.

**Figure 2 materials-13-05750-f002:**
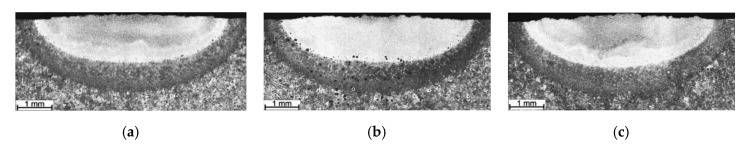
Optical macrographs of the SABs produced using a powder mixture of 80Ti-20Mo (weight ratio); SAB no. (Table 2): (**a**) TM1, (**b**) TM2, (**c**) TM3.

**Figure 3 materials-13-05750-f003:**
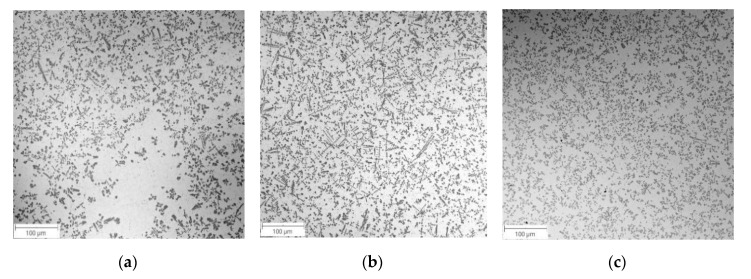
Low-magnification BSE SEM micrographs taken at the mid-section of SAB no. (**a**) T2, (**b**) TC3, (**c**) TM3 (Table 2).

**Figure 4 materials-13-05750-f004:**
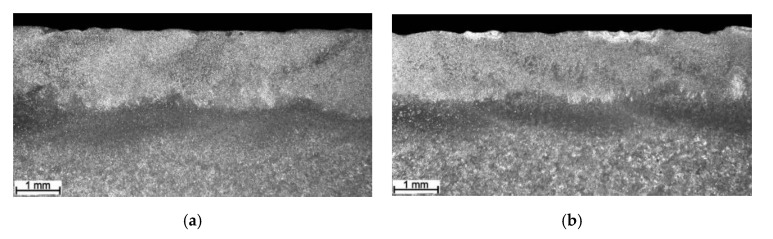
Optical macrographs of (**a**) TRC and (**b**) TRM (Table 3).

**Figure 5 materials-13-05750-f005:**
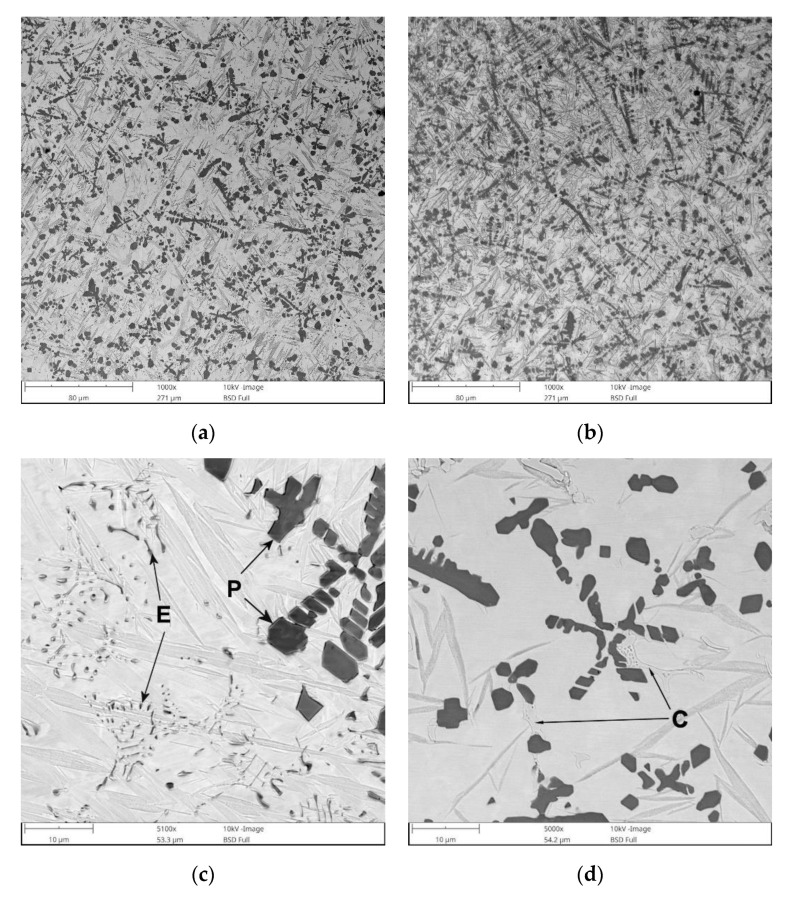
BSE SEM images showing the microstructure of TRC (Table 3): (**a**) mid-section of the layer, (**b**) the overlap boundary of the layer, (**c**) a detail from (**a**) showing the primary (*P*) and eutectic (*E*) TiC precipitates, and (**d**) a detail from (**a**) showing the distribution of eutectic-like structures formed during the last stages of solidification (**c**).

**Figure 6 materials-13-05750-f006:**
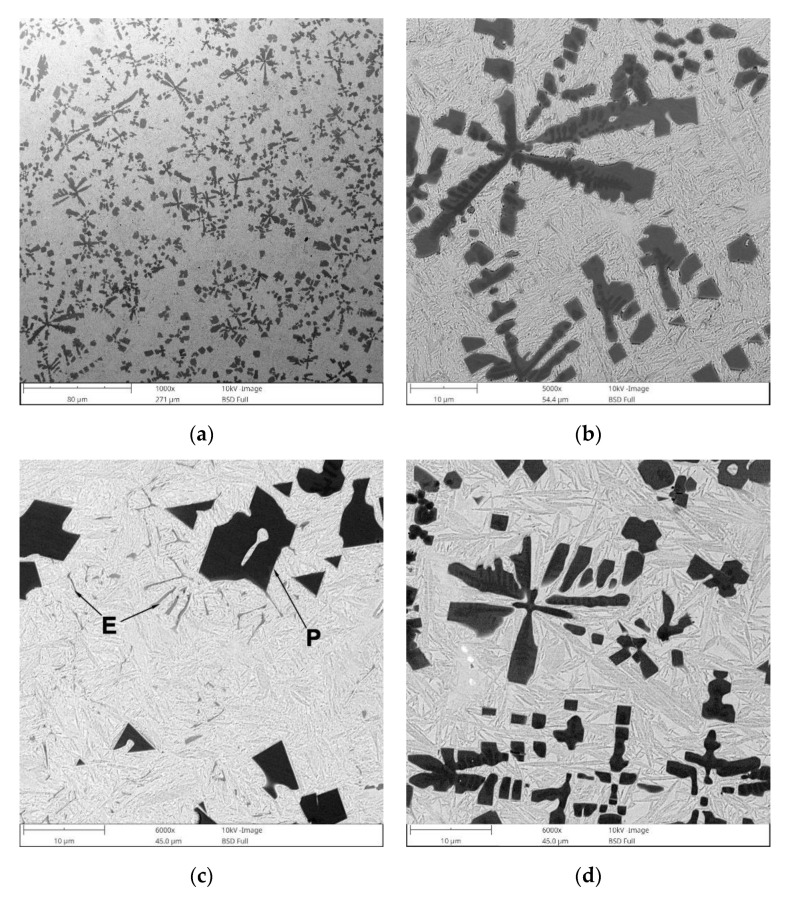
BSE SEM images showing the microstructure of the TRM (Table 3): (**a**) mid-section of the layer, (**b**) a detail from (**a**) showing typical morphology of primary TiC precipitates, (**c**) a detail from (**a**) showing a distribution of primary (*P*) and eutectic (*E*) TiC precipitates, and (**d**) image taken from the overlap boundary of the layer.

**Figure 7 materials-13-05750-f007:**
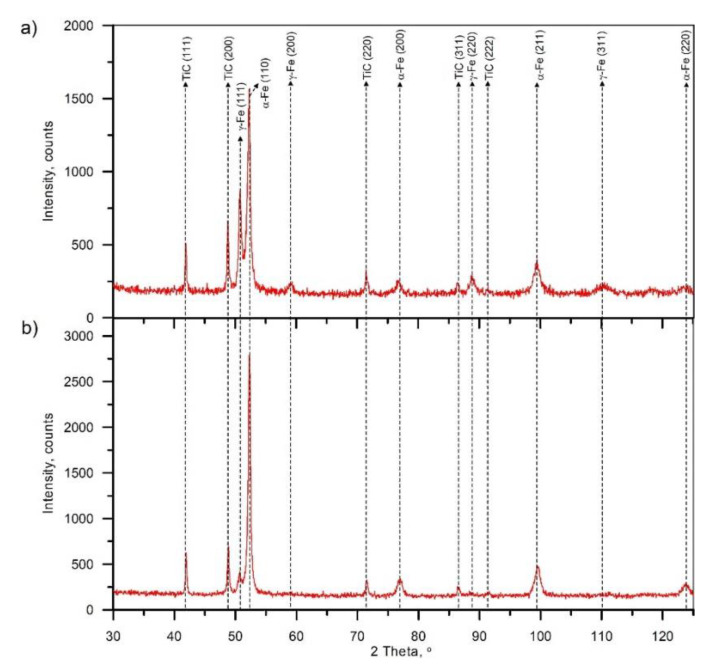
X-ray diffraction (XRD) patterns of (**a**) TRC and (**b**) TRM (Table 3).

**Figure 8 materials-13-05750-f008:**
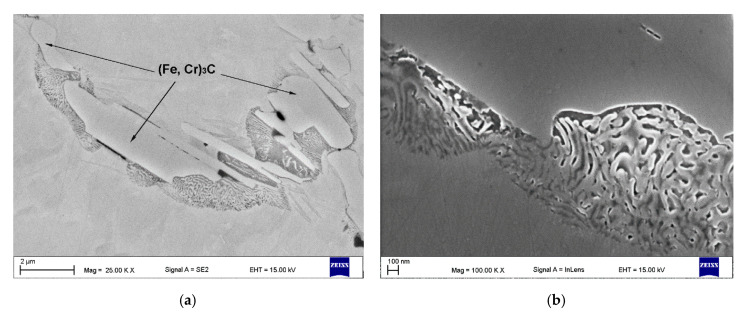
(**a**,**b**) SEM micrographs taken from the mid-section of the TRC (Table 3) showing eutectic-like structures due to the last solidification stages, (**b**) detail from (**a**), and (**c**–**g**) EDS maps of C, Si, Ti, Cr, and Fe distributions respectively, from (**a**).

**Figure 9 materials-13-05750-f009:**
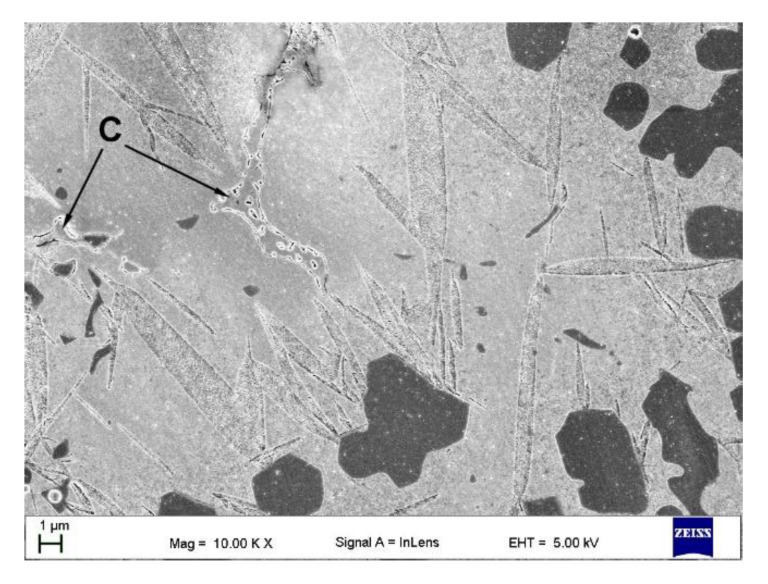
In-lens SEM micrograph taken from the mid-section of the TRM (Table 3) showing the eutectic-like structures (*C*) as a result of the last solidification stages.

**Figure 10 materials-13-05750-f010:**
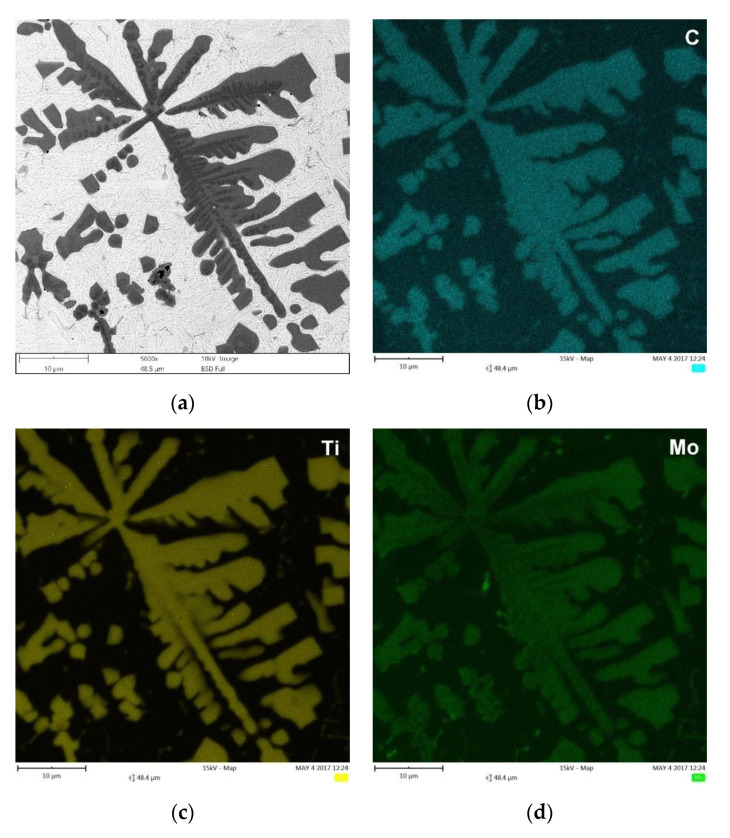
(**a**) BSE SEM micrograph taken from the mid-section of the TRM (Table 3), showing the primary (Ti, Mo)C precipitates in dendritic form and eutectic plate-like (Ti, Mo) C precipitates, and (**b**–**d**) corresponding distribution maps of C, Ti, and Mo, respectively.

**Figure 11 materials-13-05750-f011:**
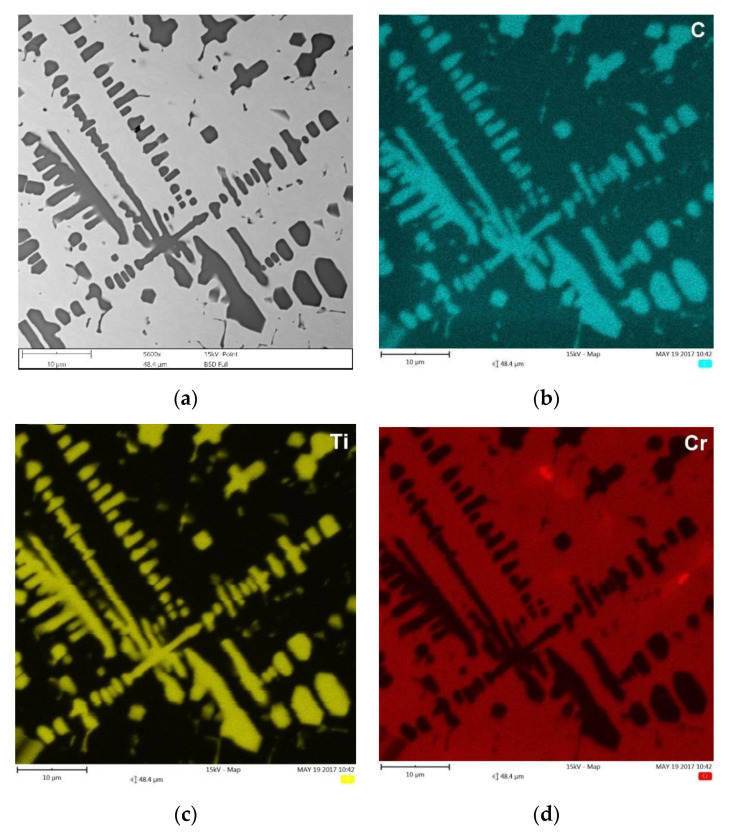
(**a**) BSE SEM micrograph taken from the mid-section of TRC (Table 3) showing the primary TiC precipitates in dendritic form and eutectic plate-like TiC precipitates, and (**b**–**d**) corresponding maps of C, Ti, and Cr distribution, respectively.

**Figure 12 materials-13-05750-f012:**
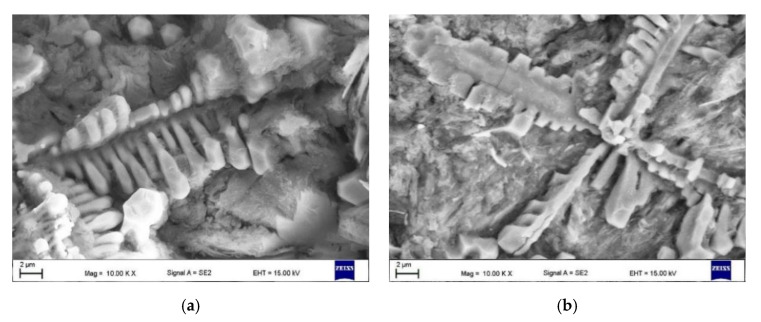
SEM micrographs of deep-etched (**a**) TRC and (**b**) TRM showing primary dendritic precipitates of TiC and (Ti, Mo)C phases, respectively (Table 3).

**Figure 13 materials-13-05750-f013:**
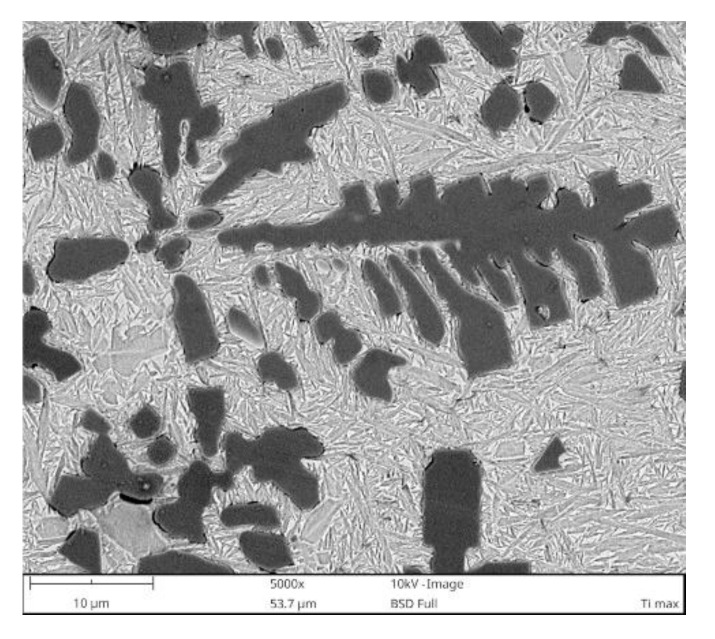
BSE SEM micrograph taken from the TRL region processed with pure Ti containing approximately 12.5 wt% Ti.

**Figure 14 materials-13-05750-f014:**
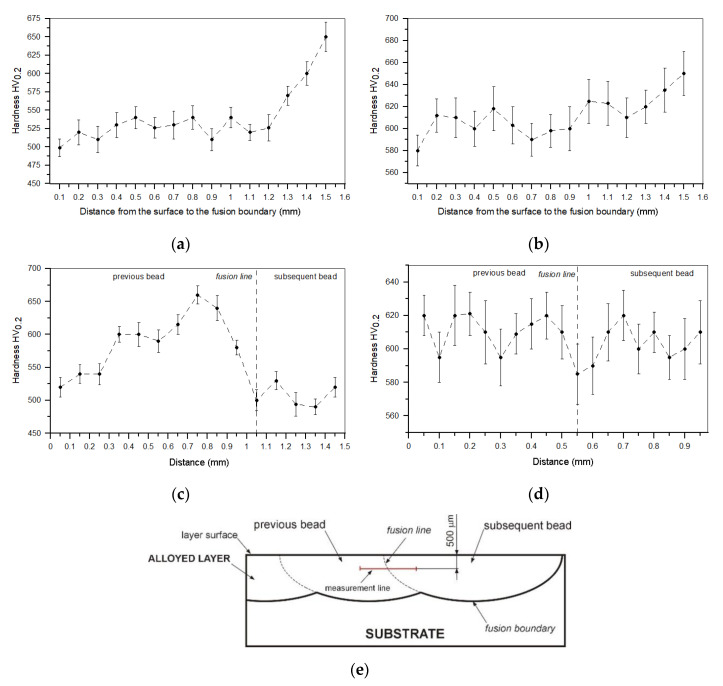
(**a**,**b**) Hardness depth profiles for TRC and TRM (Table 3), respectively. (**c**,**d**) Hardness profiles across the overlap boundary between consecutive beads in TRC and TRM, respectively. (**e**) Diagram showing a location of the measurement line in the overlap boundary.

**Table 1 materials-13-05750-t001:** Chemical composition of the used ductile cast iron (DCI) grade EN-GJS-700-2 (wt%).

C	Si	Cu	Mn	Cr	Ni	Ti	Mo	S	P	Fe
3.60	2.51	0.78	0.25	0.02	0.04	0.02	0.02	0.005	0.016	balance

**Table 2 materials-13-05750-t002:** Effect of the chemical composition of the alloying material and powder feed rate on the fusion area of the single-alloyed beads (SAB) and the concentration of alloying elements in the bead.

Processing Condition No./SAB No.	Alloying Material	Powder Feed Rate ^2^ (mg/mm)	Fusion Area of the SAB (mm^2^)	Average Ti Content, (wt%)	Average Cr Content, (wt%)	Average Mo Content, (wt%)	Quality ^3^
T1	Ti	8.0	7.34 ± 0.61	7.0 ± 0.8	−	−	U
T2		10.0	7.05 ± 0.58	8.8 ± 1.1	−	−	U
T3		11.0	6.80 ± 0.50	−	−	−	N
TC1	Ti-Cr ^1^	9.0	7.73 ± 0.63	8.4 ± 0.29	1.6 ± 0.11	−	U
TC2		11.0	7.89 ± 0.65	10.3 ± 0.38	1.9 ± 0.15	−	U
TC3		12.0	8.50 ± 0.67	11.6 ± 0.58	2.3 ± 0.24	−	U
TC4		13.0	8.42 ± 0.71	−	−	−	N
TM1	Ti-Mo ^1^	10.0	8.20 ± 0.63	7.0 ± 0.38	−	1.8 ± 0.16	U
TM2		12.5	8.09 ± 0.62	8.4 ± 0.46	−	2.3 ± 0.23	U
TM3		13.5	8.10 ± 0.64	9.9 ± 0.69	−	2.8 ± 0.32	U
TM4		14.0	8.07 ± 0.69	−	−	−	N

^1^ Powder mixtures prepared with a weight ratio of 80–20; ^2^ defined as the amount of the powder provided per unit length of the SAB; ^3^ assessed in terms of the compositional uniformity (Ti concentration): U—uniform, N—non-uniform.

**Table 3 materials-13-05750-t003:** Effect of the chemical composition of the alloying material on the microstructural parameters of the surface alloyed layers (SALs).

TRL No.	Processing Condition No. (Table 2) ^1^	α-Fe (Martensite) Fraction (wt%) ^2^	Retained Austenite Fraction (wt%) ^2^	Cementite Fraction (vol%)	TiC Fraction (vol%)
TR	T2	66.4 ± 1.1	15.3 ± 1.4	8.1 ± 2.9	15.4 ± 2.1
TRC	TC3	52.9 ± 1.6	23.7 ± 1.2	2.1 ± 0.6	20.8 ± 1.1
TRM	TM3	69.1 ± 1.4	5.7 ± 1.4	1.9 ± 0.6	21.1 ± 1.3

^1^ Overlap ratio: 30%; ^2^ based on the Rietveld refinement of X-ray diffraction (XRD) data.

**Table 4 materials-13-05750-t004:** Chemical compositions of the SALs.

TRL No.	Element (wt%)									
	C	Ti	Cr	Mo	Mn	Si	Cu	S	P	Fe
TR	3.15	9.5	0.05	0.02	0.13	1.99	0.79	0.148	0.023	balance
TRC	3.10	12.4	2.87	0.04	0.13	1.90	0.81	0.181	0.025	balance
TRM	3.07	10.5	0.06	3.15	0.14	1.85	0.76	0.177	0.023	balance

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
