# Peer review of "Effect of Chromium and Molybdenum Addition on the Microstructure of In Situ TiC-Reinforced Composite Surface Layers Fabricated on Ductile Cast Iron by Laser Alloying"

_materials, 2020, doi:10.3390/ma13245750_

Round 1

Reviewer 1 Report

This paper reported the effect of Cr and Mo addition on the microstructure of TiC-reinforced composite surface layers fabricated on ductile cast iron by laser alloying. Various bead morphologies, TiC precipitate distribution and its evolution, and the hardness of the laser surface alloyed layer were studied extensively. I believe that this study is scientifically meaningful but there are still some issues remained before going to publication. Comments and suggestions to improve the manuscript are shown below:

  1. Please state the particle size ranges of pure Ti, Cr, Mo powders prepared in this study.
  2. I believe that it is important to add the reason for the selection of Mo and Cr as additional elements into Ti and reason for the selection of weight percentage ratios of 80-20.
  3. The unit of fusion area of the SAB in Table 2 is lacking.
  4. In the previous study (Ref. 24), some cracks and voids were observed in the single bead. Please mention whether the defect formation behavior was varied or not by adding Mo and Cr elements.
  5. Line 256: Mo “phase” content ? Please clarify whether the Mo content is volume fraction or chemical composition.
  6. The results of the XRD measurement (Figure 7) are not mentioned in the manuscript although they give important data. From the XRD patterns, it can be found that Mo phase in TRM were not detected in the XRD patterns, indicating that they were almost solid solutioned in the TiC phase, as the author mentioned it from SEM analysis (Line 257). Please add some results and discussion obtained from XRD results.
  7. Not only the microstructural differences, hardness comparison of TRM and TRC (Figure 14) with TR would help to show the effect of additional elements on the mechanical properties more clearly.

Author Response

Thank you for your thorough review of my manuscript. I have corrected the manuscript considering the comments, and I provided the required explanations. 

Comment 1:  Please state the particle size ranges of pure Ti, Cr, Mo powders prepared in this study.

Response: I inserted additional information in the Experimental:

“The alloying materials were prepared using commercially-available powders of pure Ti (99% purity, particle size: 45÷70 µm, H.C. Starck), pure Cr (99% purity, particle size: 45÷70 µm, abcr GmbH), and pure Mo (99.95% purity, particle size: 5÷10 µm, abcr GmbH).”

Comment 2:  I believe that it is important to add the reason for the selection of Mo and Cr as additional elements into Ti and reason for the selection of weight percentage ratios of 80-20.

Response: I inserted the following sentences in the Experimental:

“Based on unpublished work [47], Ti-Mo and Ti-Cr powder mixtures were prepared in weight percentage ratios of 80–20. The main purpose of the additional introduction of Cr and Mo powders into the molten pool during the synthesis of TRLs was to affect the morphology and composition of the reinforcing phase (TiC precipitates) and also the phase composition of the matrix material.”

Comment 3:  The unit of fusion area of the SAB in Table 2 is lacking.

Response: I corrected Table 2 providing the suitable unit.

Comment 4:  In the previous study (Ref. 24), some cracks and voids were observed in the single bead. Please mention whether the defect formation behavior was varied or not by adding Mo and Cr elements.

Response: I inserted the following sentences in the Results and Discussion (Macrostructure Analysis):

“Regardless of the used alloying material, both SABs and SALs contained a cracks induced due to a lack of preheating the substrate. The chemistry of alloying material had a little effect on the cracking tendency of the alloyed zone. All types of SALs exhibited minor porosity and negligible tendency to a formation of microvoids.”

Comment 5: Line 256: Mo “phase” content ? Please clarify whether the Mo content is volume fraction or chemical composition.

Response: This sentence (line 256) is related to the Mo (element) concentration in the liquid in the molten pool (expressed in wt %).

Comment 6: The results of the XRD measurement (Figure 7) are not mentioned in the manuscript although they give important data. From the XRD patterns, it can be found that Mo phase in TRM were not detected in the XRD patterns, indicating that they were almost solid solutioned in the TiC phase, as the author mentioned it from SEM analysis (Line 257). Please add some results and discussion obtained from XRD results.

Response: The XRD method in the presented study was used to both qualitative and quantitative phase analysis of the layers. The results of XRD analysis are presented in Table 3. I used Rietveld refinement to estimate the fractions of the martensite and retained austenite phases in the matrix materials. I added the remark below Table 3 and the following sentences in the Results and Discussion:

2 based on the Rietveld refinement of XRD data.”

“(Fe, Cr)3C phase was not detected by the XRD due to low fraction of this phase.”

“As in the case of (Fe, Cr)3C phase in TRC, the XRD did not detect the presence of these eutectic precipitates.”

Comment 7: Not only the microstructural differences, hardness comparison of TRM and TRC (Figure 14) with TR would help to show the effect of additional elements on the mechanical properties more clearly.

Response: TRM and TRC layers contained similar volume fractions of TiC precipitates and in consequence similar (negligible) contents of cementite. Moreover, the achieved concentrations of MC carbide (TiC) forming elements in the above layers were close to optimal (which was predicted by the thermodynamic calculations presented in [24]). It enabled a direct comparison of their characteristics. As a result, the manuscript was focused on the microstructural and hardness analysis of these two types of TRLs.

Reviewer 2 Report

Complex microstructure of SAB described in extensive detail.  The pure Ti was done in a previous publication, but puts the burden on the reader to go and find out what these results were for comparison to current study.  

Editing figures is needed so that regions of interest (ROIs) described in the text can be found in figures.  The EDS elemental maps need to be labeled on the map, not just in the figure caption. It is difficult to check if Mo or Cr is on dendrite or matrix.  There is a lot of different variation in details from the samples that makes the results hard to follow as presented.

The linescan EDS data is very noisy and does not show clearly the composition of the feature indicated. In the text it states what the feature is but the data does not do a convincing job of showing data to back up the claim.  When presenting quantitative data in the paper, repeatedly, one needs to include the actual kV and software used for quant analysis. Was a ZAF correction and background subtraction used?  Was the LECO data being used or EDS quant? It was very confusing since only a reference was given that cited the SEM parameters.  Needs to be directly stated in experimental sections or else a lot of the conclusions become invalid since methods of arriving at conclusions becomes suspect.

The hardness data was difficult to understand where the data was acquired from. What is "adjacent bead"?  It may be best to illustrate on xsec micrograph where hardness data was being collected from. Also, all of the hardness data could be graphed on a single graph for easier comparison. It seemed error bars were large. How many times was the hardness data collected? Was this just from a single profile?

Author Response

Thank you for your thorough review of my manuscript. I have corrected the manuscript considering the comments, and I provided the required explanations. 

Comment 1: Editing figures is needed so that regions of interest (ROIs) described in the text can be found in figures.  The EDS elemental maps need to be labeled on the map, not just in the figure caption. It is difficult to check if Mo or Cr is on dendrite or matrix.  There is a lot of different variation in details from the samples that makes the results hard to follow as presented.

Response: I labelled EDS elemental maps in Figures 8, 10 and 11. Moreover, I inserted proper labels to Figures 5, 8a and 9 to indicate ROIs.

Comment 2: The linescan EDS data is very noisy and does not show clearly the composition of the feature indicated. In the text it states what the feature is but the data does not do a convincing job of showing data to back up the claim.

Response: I removed the linescan EDS data from the manuscript. Instead, I inserted a new image showing the morphology of the analysed precipitates (with notably better quality than that previously used). Additionally, I modified the description of these eutectic regions in the text:

“The typical eutectic-like structure formed during the last stages of solidification in TRM is presented in Figure 9. The fraction of these eutectic regions in the layer was estimated to be 1.9 ± 0.6 vol %. As in the case of (Fe, Cr)3C phase in TRC, the XRD did not detect the presence of these eutectic precipitates. However, considering the morphology of the above eutectic-like structure and the chemistry of the layer one can assume that it contained mainly the cementite phase.”

The fact is that, due to the small size of these precipitates, to provide reliable data on their composition, the additional TEM investigations are needed. However, it should be pointed out that, due to their low fraction in the layer (< 2 vol %), they have a negligible effect on the overall properties of the investigated TiC-reinforced composite layers.  As a result, a detailed characterization of these precipitates is not very important for the presented manuscript. However, considering the morphology of these eutectic precipitates and the chemistry of the analysed alloy system (possible solidification paths) it is reasonable to assume that the main constituent of these eutectic regions is cementite phase (enriched in Mo).

Comment 3: When presenting quantitative data in the paper, repeatedly, one needs to include the actual kV and software used for quant analysis. Was a ZAF correction and background subtraction used?  Was the LECO data being used or EDS quant? It was very confusing since only a reference was given that cited the SEM parameters.  Needs to be directly stated in experimental sections or else a lot of the conclusions become invalid since methods of arriving at conclusions becomes suspect.

Response: I modified the Experimental:

The microstructure of TRLs was characterized by scanning electron microscopy (SEM) with energy-dispersive spectroscopy (EDS) analysis and X-ray diffraction (XRD). EDS measurements were performed at an accelerating voltage of 15 kV. EDAX Genesis software (ver. 6.0, EDAX Inc., Mahwah, NJ, USA) was used for the collection and analysis of EDS data. The quantitative EDS analysis was made to estimate the concentration of alloying material in the matrix and also to confirm the compositional homogeneity of the alloyed layers. Additionally, chemical analysis of the SM and SALs was performed on ground surface specimens using a LECO GDS 500A optical emission spectrometer (LECO Corporation, St. Joseph, Michigan, USA). The C concentration in the SM was determined using a LECO CS125 Carbon Analyzer (LECO Corporation, St. Joseph, Michigan, USA). The procedures and equipment used during microstructural characterization are described in greater detail elsewhere [24,25]. “

The EDAX Genesis software uses the ZAF correction method. The LECO was used to measure the overall chemical composition of the layers (on the top surface of the layer).

Comment 4: The hardness data was difficult to understand where the data was acquired from. What is "adjacent bead"?  It may be best to illustrate on xsec micrograph where hardness data was being collected from. Also, all of the hardness data could be graphed on a single graph for easier comparison. It seemed error bars were large. How many times was the hardness data collected? Was this just from a single profile?

Response: I inserted a scheme of hardness measurements in Figure 14e. The relatively large error bars on the hardness profiles are typical for the particle reinforced metal matrix composite materials (MMC). Each point on the hardness profile constitutes the mean value and the standard deviation from three measurements – in other words, each hardness profile was constructed from three profiles.

Reviewer 3 Report

The authors report a study on the effect of Cr and Mo addition on the microstructure of in situ TiC reinforced cast iron laser alloying coating.

The paper is relevant, well founded and discussed, and of interest to the audience of this journal. The introduction and literature review provides the necessary background information for understanding the methodology, which is appropriate and applied properly.

However, there are some observations, and modification are recommended before publication:

1.Please present the particle size of the powders in the text, which directly affects the feasibility of automatic powder feeding for laser surface alloying.

  1. Besides the laser power, scanning speed and powder feeding rate had been mentioned in the paper, the laser beam diameter, overlap ratio and defocusing are also important parameters, please present them in the text.
  2. It can be found from Fig. 1 and Fig. 2 that with the increase of powder feeding rate, the penetration of laser pool first decreases and then increases, why?
  3. By comparing Fig. 1 and Fig. 2, why is the penetration of coating with Ti-Mo powder less than that of the coating with Ti-Cr powder, why?

 Please clarify it.

  1. From Fig. 1 and Fig. 2, it can be found that the number of pores in the coating for Ti-Cr powder increases with the increasing of powder feeding rate, while that of Ti-Mo powder is opposite. Please clarify it.
  2. It can be seen from Fig.3 that dendrites of reinforcements become coarse after adding Cr element in the LA coating, however, dendrites of reinforcements become fine after adding Mo element. Please clarify it.

7. Some of the discussions are poor and should be proved with evidence. If this is not the case, some references should be incited. Such as, ” the fusion zone shape of SABs produced with Cr addition indicates that the coefficient of the surface tension temperature on the molten pool surface was positive, producing fluid flow inward  along the molten pool surface.”

8.Comparing of distribution and size of reinforcement (as shown in Fig. 3), why different powder feeding rate is used. What is the distribution of reinforcing phase in the coating at same powder feeding rate?

9. In Fig. 4, the coatings are prepared at what kind of powder feeding rate conditions, please present it in the text.

10. The quality of Figure 9 is poor, please replace it.

11. The so-called (Fe, Mo) 3C phase can not be clearly found from line distribution of elements in Fig. 9, please present TEM analysis or atomic ratio of element in the phases.

12. (Fe,Mo)3C and (Ti, Mo) C multiple carbide is found in the coating (Fig. 10), but it is not observed in the XRD, why? please clarify it in the text.

13. Based on the Fig. 14(a) and (b), it can be found that the hardness of the TRC coating shows a gradient distribution, and the hardness of the coating increases with the increasing of distance from surface. However, the hardness of the coating with Mo element is slight evenly distributed. In fact, the density of molybdenum carbide is higher than that of chromium carbide. Why the hardness distribution of TRM coating showed much uneven than that of TRC coating?

14. Conclusions should be reorganized and concise.

Author Response

Thank you for your thorough review of my manuscript. I have corrected the manuscript considering the comments, and I provided the required explanations. 

Comment 1:  Please present the particle size of the powders in the text, which directly affects the feasibility of automatic powder feeding for laser surface alloying.

Response: I inserted additional information in the Experimental:

“The alloying materials were prepared using commercially-available powders of pure Ti (99% purity, particle size: 45÷70 µm, H.C. Starck), pure Cr (99% purity, particle size: 45÷70 µm, abcr GmbH), and pure Mo (99.95% purity, particle size: 5÷10 µm, abcr GmbH).”

Comment 2:  Besides the laser power, scanning speed and powder feeding rate had been mentioned in the paper, the laser beam diameter, overlap ratio and defocusing are also important parameters, please present them in the text.

Response: I inserted the following sentences in the Experimental:

“A laser beam spot size (rectangular in shape) was 1.5 x 6.6 mm. During all trials, the laser spot was located on the top surface of SM, and the fast-axis of the laser beam was set parallel to the scanning direction.”

“For microstructure analysis of the TRLs synthesized at different melt compositions, surface alloyed layers (SAL) were produced via a multi-pass overlapping alloying process with an overlap ratio of 30% (Tables 3 and 4).”

Comments 3 and 4:  It can be found from Fig. 1 and Fig. 2 that with the increase of powder feeding rate, the penetration of laser pool first decreases and then increases, why?

By comparing Fig. 1 and Fig. 2, why is the penetration of coating with Ti-Mo powder less than that of the coating with Ti-Cr powder, why? Please clarify it.

Response: The shape and dimensions of the fusion zone are directly affected by the heat and mass transfer in the molten pool - the intensity and pattern of the fluid flow (Marangoni convection) in the molten pool. The data indicate that both Mo and Cr powders enhanced intensity of convection in the molten pool. The Cr powder had especially beneficial effect on the intensity of the convective flow of liquid in the molten pool. As a result, SABs produced with Ti-Cr mixture had a larger fusion zone area than that produced with Ti-Mo mixture. I measured the fusion zone area – in the case of Ti-Cr mixture, the fusion zone area increased with increasing powder feed rate (in the optimal range of powder feed rate). Additionally, in the case of Cr addition, the shape of the fusion zone suggested that the fluid flow was directed into the molten pool centre along the pool surface (positive temperature coefficient of the surface tension) – providing the deeper molten pool. In the case of Ti-Mo mixture, the fusion zone area was, in general, not affected by the powder feed rate (in the optimal range of powderfeed rate). I think that it is reasonable to assume that the notably higher melting point of Mo than that of Cr may have affected the thermal conditions in the molten pool and as a consequence fusion zone area.

Comment 5: From Fig. 1 and Fig. 2, it can be found that the number of pores in the coating for Ti-Cr powder increases with the increasing of powder feeding rate, while that of Ti-Mo powder is opposite. Please clarify it.

Response: Generally, all types of alloyed beads/layers were free of porosity. Macrographs in Figures 1 and 2 reveal that dissolution of graphite nodules (directly in the area adjacent into the fusion boundary) was not complete.

Comment 6: It can be seen from Fig.3 that dendrites of reinforcements become coarse after adding Cr element in the LA coating, however, dendrites of reinforcements become fine after adding Mo element. Please clarify it.

Response: Figure 3 shows, in general, the distribution of TiC precipitates throughout the SABs. The morphology of TiC-type reinforcing particles in the investigated layers can be analysed using Figures 5, 6, 10, 11, 12 and 13.

Comment 7: Some of the discussions are poor and should be proved with evidence. If this is not the case, some references should be incited. Such as, ” the fusion zone shape of SABs produced with Cr addition indicates that the coefficient of the surface tension temperature on the molten pool surface was positive, producing fluid flow inward  along the molten pool surface.”

Response: I inserted suitable references into the above sentence:

“The fusion zone shape of SABs produced with Cr addition indicates that the coefficient of the surface tension temperature on the molten pool surface was positive, producing fluid flow inward along the molten pool surface [32-34].”

Comment 8: Comparing of distribution and size of reinforcement (as shown in Fig. 3), why different powder feeding rate is used. What is the distribution of reinforcing phase in the coating at same powder feeding rate?

Response: The difference in the powder feed rates resulted from the ability to produce uniformly alloyed zone (uniform distribution of alloying material throughout the alloyed zone), which, in turn, was dependent on the intensity of the molten pool convection (for the given alloying material chemistry). Moreover, the powder feed rates (expressed in mg/mm) depended on the densities of the used powders. In the optimal range of powder feed rate for the given alloying powder, TiC particles were uniformly distributed throughout the alloyed zone. The different powder feed rates leading to various contents of TiC forming elements (Ti and Mo) in the melt resulted in the different TiC fractions in the layers.

Comment 9: In Fig. 4, the coatings are prepared at what kind of powder feeding rate conditions, please present it in the text.

Response: The processing conditions used to fabricate the alloyed layers (showed in Figure 4) are presented in Table 3.

Comments 10 and 11: The quality of Figure 9 is poor, please replace it.

The so-called (Fe, Mo) 3C phase can not be clearly found from line distribution of elements in Fig. 9, please present TEM analysis or atomic ratio of element in the phases.

Response: I removed the linescan EDS data from the manuscript. Instead, I inserted a new image showing the morphology of the analysed precipitates (with notably better quality than that previously used). Additionally, I modified the description of these eutectic regions in the text:

“The typical eutectic-like structure formed during the last stages of solidification in TRM is presented in Figure 9. The fraction of these eutectic regions in the layer was estimated to be 1.9 ± 0.6 vol %. As in the case of (Fe, Cr)3C phase in TRC, the XRD did not detect the presence of these eutectic precipitates. However, considering the morphology of the above eutectic-like structure and the chemistry of the layer one can assume that it contained mainly the cementite phase.”

The fact is that, due to the small size of these precipitates, to provide reliable data on their composition, the additional TEM investigations are needed. However, it should be pointed out that, due to their low fraction in the layer (< 2 vol %), they have a negligible effect on the overall properties of the investigated TiC-reinforced composite layers.  As a result, a detailed characterization of these precipitates is not very important for the presented manuscript. However, considering the morphology of these eutectic precipitates and the chemistry of the analysed alloy system (possible solidification paths) it is reasonable to assume that the main constituent of these eutectic regions is cementite phase (enriched in Mo).

Comment 12: (Fe,Mo)3C and (Ti, Mo) C multiple carbide is found in the coating (Fig. 10), but it is not observed in the XRD, why? please clarify it in the text.

Response: (Ti, Mo)C phase was detected by XRD as TiC phase (the same crystal structure). To explain the absence of the eutectic precipitates (formed during the last stages of solidification in both TRC and TRM) on the XRD pattern, I inserted the following sentences in the Results and Discussion:

(Fe, Cr)3C phase was not detected by the XRD due to low fraction of this phase.”

“As in the case of (Fe, Cr)3C phase in TRC, the XRD did not detect the presence of these eutectic precipitates. However, considering the morphology of the above eutectic-like structure and the chemistry of the layer one can assume that it contained mainly the cementite phase.”

Comment 13: Based on the Fig. 14(a) and (b), it can be found that the hardness of the TRC coating shows a gradient distribution, and the hardness of the coating increases with the increasing of distance from surface. However, the hardness of the coating with Mo element is slight evenly distributed. In fact, the density of molybdenum carbide is higher than that of chromium carbide. Why the hardness distribution of TRM coating showed much uneven than that of TRC coating?

Response: The gradient distribution of hardness across the layer thickness results from the distribution of alloying elements in the layer, and it is associated with the mechanism of the surface alloying process (mass transport in the molten pool). Directly at the fusion boundary, there is a zone having a gradient distribution of the alloying material (a concentration of the alloying material decreases with decreasing distance to the fusion boundary). The lower concentration of MC carbide (TiC) forming elements in this zone (in the case of the investigated alloy system) leads to a higher fraction of cementite phase and higher hardness. The TRC layer had higher fusion depth, and the above zone was slightly thicker.

Comment 14: Conclusions should be reorganized and concise.

Response: I would like not to change a form of Conclusions – this section is relatively short and presents the main findings of the study.

Reviewer 4 Report

In this research, the authors analyzed the effect of Mo and Cr addition on the evolved nonstructural attributes in a laser surface alloying process on a DCI substrate.

The article is written well and provides a useful understanding of the phenomena for future engineering applications.

It would be great if the author can improve the quality of the figures and tables for an even better presentation of the results before the article is published.

  • Please add mid-table separation lines in Table 2 After T3 and TC4.
  • Scales in most of the provided micrographs are not clearly visible, either too small or in the same color as the background.
  • It would be great if the author can construct and label the micrographs by pointing out the features that are discussed in the article body. This helps readers better understand the observations.

Use the past tense to report what happened in the past: what you did, what someone reported, what happened in an experiment, and so on. Use the present tense to express general truths, such as conclusions (drawn by you or by others) and atemporal facts (including information about what the paper does or covers). Please avoid using I, you, we i.e. on Page 2, Line 51.

As there are many acronyms and symbols used in the article. Please add nomenclature before references with a list of acronyms and symbols used in the article. 

Please carefully prepare the revised draft of the article by following the journal template.

After these modifications, I believe the article will be in much better shape for publication

Author Response

Thank you for your thorough review of my manuscript. I have corrected the manuscript considering the comments.

Comment 1:  Please add mid-table separation lines in Table 2 After T3 and TC4.

Response: I corrected Table 2 according to the comment.

Comment 2: Scales in most of the provided micrographs are not clearly visible, either too small or in the same color as the background.

Response: I improved scales in Figures 8, 9, 10a, 11a, 12.

Comment 3: It would be great if the author can construct and label the micrographs by pointing out the features that are discussed in the article body. This helps readers better understand the observations.

Response: I inserted proper labels to Figures 5, 8a and 9.

Comment 4: Use the past tense to report what happened in the past: what you did, what someone reported, what happened in an experiment, and so on. Use the present tense to express general truths, such as conclusions (drawn by you or by others) and atemporal facts (including information about what the paper does or covers). Please avoid using I, you, we i.e. on Page 2, Line 51.

Response: I tried to improve the manuscript in these terms. (Please note that the manuscript has been proofread by a native speaker of English who holds a PhD in Chemistry). Among others, I corrected the following sentence:

“Previous studies [23-25] were undertaken to determine the feasibility of the in situ synthesis of TRLs on the DCI substrate via LA with pure Ti powder.”

“It is reasonable to assume that the lamellar precipitates enriched in Cr were (Fe, Cr)3C phase.”

Comment 5: As there are many acronyms and symbols used in the article. Please add nomenclature before references with a list of acronyms and symbols used in the article. 

Response: I added Appendix A with Table A1 presenting abbreviations used in the manuscript. Moreover, I inserted the following sentence in the Experimental:

“Abbreviations used in the manuscript are also listed in Appendix A (Table A1).”

Round 2

Reviewer 3 Report

1.Ref. [11], please change Al2O3 as Al2O3;

2.Ref. [12], please change  Al2O3 and TiO2 as Al2O3 and TiO2, respectively.

3.P2 line 69-70, what, "particle size: 45÷70 µm" and "particle size: 5÷10 µm", is the meaning of this? Is it 45~70 µm and 5~10 µm or 0.64 µm and 0.5 µm? Please clarify it

Author Response

Thank you for your deep and thorough review. I have corrected my paper taking into account all comments.

Comment 1:   “Ref. [11], please change Al2O3 as Al2O3;”

Response: I corrected it.

Comment 2:   “Ref. [12], please change  Al2O3 and TiO2 as Al2O3 and TiO2, respectively.”

Response: I corrected it.

Comment 3:  “P2 line 69-70, what, "particle size: 45÷70 µm" and "particle size: 5÷10 µm", is the meaning of this? Is it 45~70 µm and 5~10 µm or 0.64 µm and 0.5 µm? Please clarify it”

Response: It is the particle size range (provided by the manufacturer). I modified this sentence as follow:

“The alloying materials were prepared using commercially-available powders of pure Ti (99% purity, the particle size range of 45 to 70 µm, H.C. Starck), pure Cr (99% purity, the particle size range of 45 to 70 µm, abcr GmbH), and pure Mo (99.95% purity, the particle size range of 5 to 10 µm, abcr GmbH).”